# Barriers and Facilitators of Health and Well-Being in Informal Caregivers of Dementia Patients: A Qualitative Study

**DOI:** 10.3390/ijerph20054328

**Published:** 2023-02-28

**Authors:** Sally C. Duplantier, Francesca A. Williamson

**Affiliations:** 1Department of Counseling and Educational Psychology, School of Education, Indiana University, 201 Rose Avenue, Bloomington, IN 47405, USA; 2Department of Pediatrics, School of Medicine, Indiana University, 340 W. 10th Street, Fairbanks Hall, Suite 6200, Indianapolis, IN 46202, USA

**Keywords:** informal caregivers, family caregivers, Alzheimer’s, dementia, subjective burden, objective burden, grief, anticipatory grief, pre-death grief, health, well-being

## Abstract

Background: Given the dramatic projected increase in Alzheimer’s disease globally and the increased risk of morbidity and mortality for family caregivers of these patients, there is an urgent need to provide more targeted, timely resources to support the health and well-being of these informal caregivers. Few studies have investigated the barriers to health and well-being and potential strategies to facilitate better self-care from the unique perspective of the caregivers themselves. Purpose: This qualitative study aimed to identify barriers and facilitators to health and well-being for informal caregivers of family members with Alzheimer’s. Method: We conducted semi-structured interviews with eight informal caregivers, including daughters, wives, and one husband, ages 32 to 83. Using Reflexive Thematic Analysis, we identified three main themes and subthemes across caregivers’ experiences. Findings: We found that caregivers (1) prioritized mental and social well-being over physical health or health behaviors; (2) characterized the subjective burden of caregiving as a “mantle of responsibility” that could not easily be shed due to the complex subjective burden of loss, grief, guilt, resentment, isolation, loneliness, and lack of agency; (3) sought to be recognized as “additional patients”, instead of being viewed as invisible patients, with support services tailored to their life stage and challenges. Conclusions: The findings suggest that the subjective burden of strain experienced by family caregivers of Alzheimer’s patients has a profound impact on their health and well-being, even more so than the objective burden of strain that is the result of their day-to-day caregiving activities.

## 1. Introduction

Worldwide, Alzheimer’s disease affects 55 million people and is expected to reach 139 million people by 2050, at a cost of nearly $1.1 trillion [1]. An estimated 83% of these Alzheimer’s patients are cared for in their homes by more than 16 million Americans who provide over 16 billion hours of unpaid care, valued at more than $271 billion [2]. These informal caregivers, primarily family members, are often called “invisible second patients” because they experience higher levels of physiological and psychological burden, social isolation, and financial hardship when compared to other caregivers and non-caregivers [3,4,5].

The increased risk of morbidity and mortality these caregivers face is due to caring for the unique needs of people with impaired memory, cognition, and judgment. Compared with other caregivers, those tending to Alzheimer’s patients provide care that is more physically demanding, time consuming, and stressful. They engage in the most difficult type of care (e.g., bathing and dealing with incontinence); they often provide near constant care (40+ hours a week); and they do this for long periods of time (more than five years) [6]. Compared with other caregivers, the informal caregivers of people with Alzheimer’s and other dementias are at greater risk of anxiety and depression [7]. Their levels of psychological stress are significantly higher than other caregivers, and their levels of self-efficacy, subjective well-being, and physical health are significantly lower [3]. The burden of caregiving also impacts the caregivers’ ability to work and secure their own financial future. Two-thirds of working Alzheimer caregivers missed work because of their caregiver responsibilities; 14% opted out of the workforce; 13% reduced their work hours; and 8% turned down a promotion [6].

Thus, Alzheimer’s disease and related dementias have a devastating effect on the health of informal caregivers as well as the patients with the disease. Nearly 75% of these caregivers’ report concern about maintaining their own health and well-being and 1 in 3 caregivers say their personal health has declined since becoming a caregiver. More than 25% of these caregivers postpone their own health maintenance activities because of their caregiving responsibilities; approximately 40% of them suffer from depression [8]. These caregivers experience higher levels of chronic disease (cardiovascular disease, diabetes, arthritis, ulcers, anemia), are less likely to participate in preventative health behaviors such as healthy eating and exercise and are more likely to participate in unhealthy behaviors such as smoking and alcohol consumption [3,9].

Numerous studies have assessed the impact of caregiving on caregivers’ health, but few have analyzed its impact on caregivers’ self-care or with a deep understanding of that impact from the caregiver perspective. The purpose of this study is to better understand the barriers to health and well-being and potential strategies to promote better self-care from the unique perspective of the caregivers themselves. It reflects the lived experiences of husbands, wives, and daughters who care for loved ones with dementia for years or even decades. The emphasis of this research is on the subjective burden of care: the caregivers’ perception of the physical and emotional impact of their caregiving role [10]. The subjective burden of care is under-represented in the literature when compared to the objective burden of care (e.g., the time, physical activities, and finances devoted to care [11,12]). The findings of this study show ways in which grief, loss, regret, guilt, loneliness, and isolation lead to a lack of agency and impact caregivers’ ability to prioritize their own health. This study also demonstrates the need for these informal caregivers to be seen by the healthcare system as “additional patients”, rather than “invisible patients” [3], enabling them to better prioritize their own well-being.

Given the projected exponential increase in Alzheimer’s disease and the extraordinary impact on informal caregivers, this study is significant and timely. The purpose of this study was to identify barriers and facilitators to caregivers’ health and well-being and presents findings that can inform the offerings of the many governmental, private, and non-profit organizations that support Alzheimer’s patients and their families.

## 2. Materials and Methods

### 2.1. Study Design

We used a qualitative research design [13] and conducted 60-min telephone interviews to collect data about participants’ caregiving experiences and the perceived impact on their health and well-being. Interviews provide insight into the perception, meaning, and motivation of behaviors from the point of view of the caregivers [14].

### 2.2. Participants

Participants were recruited using purposeful sampling [13] to identify people who could answer questions about their health and well-being based on their lived experience as a caregiver for a family member with dementia. To identify potential participants, we emailed study information to three organizations that provide educational resources, respite care, and combined patient/caregiver activities for Alzheimer’s patients and their families. Inclusion criteria were adults aged 18 and older who have cared for a family member with Alzheimer’s or another form of dementia for at least one year.

Twelve individuals responded to the invitation to participate in the study. A total of ten family caregivers were interviewed. Two family caregivers who were interviewed were subsequently excluded because they did not meet the inclusion criteria (e.g., their family members had brain injuries but not dementia, or the caregiver was in the role for less than one year).

### 2.3. Study Procedure

The Indiana University Institutional Review Board (IRB) reviewed and approved this study in the Exempt category. All participants provided informed consent for their interviews to be used for research. Each participant was assigned an identifier to protect their anonymity.

At the start of the study, organizations that provide support to Alzheimer’s patients and caregivers were contacted with information about the study purpose and design. Individuals within these organizations were provided with an email message about the study that they could share with caregivers in their community. These caregivers could then contact the lead researcher, SD, by email to indicate interest. After indicating interest, caregivers received a study information sheet to provide them with an understanding of the purpose and procedures of the study and allow them to provide informed consent. If caregivers chose to participate, they received a list of the interview questions, and a 60-min telephone interview was scheduled using a phone number they provided. Between October 2022 and December 2022, we conducted eight (8) interviews with caregivers who met the inclusion criteria. All interviews were audio recorded.

The interviews were based on a semi-structured interview guide submitted and approved by the Indiana University IRB. To develop the interview protocol, the lead researcher (SD) reviewed the literature to identify key concepts and topics of interest related to caregiver health and well-being. A draft of the protocol was piloted with two initial participants and refined considering additional concerns that arose during the interview (two questions were clarified to distinguish between general concepts of health and well-being and the caregiver’s personal health and well-being practices). The final protocol included three sections and 10 questions with eight probing prompts, and follow-up emails by the lead researcher to clarify comments. Example questions from the final protocol were: When you think about health and well-being, what comes to mind?If I say that someone “takes care of their own health and well-being”, what does this mean to you?Has caring for a family member with dementia affected your ability to take care of your own health and well-being? If so, how?What would you need to make your own health and well-being more of a priority?

All interviews were conducted by the lead researcher, SD. Interviews began with a recap of the study purpose and a reminder that participants could pass on any questions they did not want to answer. The remainder of the interview focused on three primary areas. The first part of the interview began with a “grand tour” question [13] that helps participants respond expansively. The grand tour question asked was “Tell me a little bit about yourself and the person you care for”. The second part of the interview focused on the caregivers’ interpretation of “health and well-being”, including the ways in which their own health and well-being had changed because of their responsibilities. The third part of the interview focused on the resources, tools, and support caregivers believed would help them (and others in similar roles) better prioritize their own health and well-being.

### 2.4. Data Analysis

All interviews were recorded via Rev.com (accessed on 12 October 2022) and transcribed verbatim using Rev.com’s artificial intelligence transcription feature. The transcripts were then reviewed and lightly edited to produce clean verbatim transcripts. For the data analysis, we conducted a reflexive thematic analysis to identify patterns across discussion topics and across participants’ experiences as caregivers [15]. Reflexive thematic analysis (RTA) is a process which recognizes researcher subjectivity in the coding and theme development because the researchers are actively involved in the knowledge production. RTA is useful for analyzing the perspectives of different research participants, noting similarities and differences, and generating unanticipated insights.

For the coding and analysis, we adapted the Bingham and Witkowsky’s 5-Cycle Process [16] that includes both deductive and inductive analytical processes. To create the initial codes and generate themes, we drew upon the principles of design thinking [17]. This is a highly iterative process that is grounded in a deep and empathetic understanding of the participants and is consistent with “cognitive empathy”, a cornerstone of qualitative research [14]. After becoming familiar with the transcripts, SD and FW reviewed initial interpretations of core topics and the shared and varied features of participants’ experiences (Cycle 2). During this cycle, we attempted to achieve the research quality criteria of heterogeneity by seeking to illuminate variation and distinctiveness within and across participants’ reports of their experiences [14]. Next, we analyzed segments of the transcripts related to the subjective burden experienced by informal caregivers relevant to the research question (their interpretation of the objective burden on their well-being) (Cycle 3). The themes were further refined through analysis, synthesis, and feedback cycles between November and December 2022 (Cycle 4). In the final stage of analysis, we reviewed the literature related to the health and well-being of informal caregivers to identify similarities and differences between our findings and relevant published research (Cycle 5).

### 2.5. Reflectivity/Positionality Statement

We approached this research from an interpretive paradigm [18], recognizing that our findings would be filtered through two lenses: first, the caregivers’ perspective, that was colored by their own interpretation of remembered experiences, and second, by our perspective as researchers. To minimize the effect of our own thinking, research questions and probes were designed to be as neutral and open-ended as possible to avoid “leading the witness” with our point of view. We also engaged in multiple reflexivity practices, such as memoing and debriefing, noted variability across participants’ narratives, and identified instances that differed from the general patterns to identify instances that did not align with our expectations [19].

We also adopted a desire-based research approach versus a damage-based research approach [20]. Desire-based research looks at the complexity and contradiction of a lived experience versus simply focusing on the challenges. Our goal was to listen for the ambiguity in the caregivers’ experience, that typically included both negatives and positives in their experience.

## 3. Results

Results are presented as follows. First, socioeconomic data about the participants is provided. Then, we provide an overview of three main themes with related subthemes (Table 1). The themes and subthemes address (1) how participants defined health and well-being; (2) perceived barriers to health and well-being; and (3) strategies caregivers perceived would enable them to better prioritize their own health and well-being. Results are supported by verbatim comments from the participant interviews, along with participant IDs and relationships to the Alzheimer’s disease patient.

### 3.1. Socieodemographic Data

The eight participants who met the inclusion criteria for this study were primarily female (N = 7) and ranged in age from 32 to 83. They represented daughters (N = 4), wives (N = 3), and husbands (N = 1) of the patients. Caregiver work status was a mix between full-time work (N = 5) and retirement (N = 3). Most participants (N = 7) were married with children. All patients they cared for had Alzheimer’s disease, although one patient had Benson’s Syndrome, a rare visual variant of Alzheimer’s disease. Patients aged in range from 61 to 83, and one patient was recently deceased at age 70. Caregivers had performed their roles for many years (2.5 to 12 years). Most caregivers (N = 6) provided care in their own home, although two of these caregivers had moved their parents to residential facilities (see Table 2).

### 3.2. Health and Well-Being as Defined by Caregivers

To establish an understanding of “health and well-being” from the caregivers’ perspective, we asked participants to provide their personal definition of these two terms. While answers included health behaviors such as eating well, exercising, getting sufficient sleep, and going to the doctor, caregivers consistently prioritized mental and social well-being and de-prioritized other health behaviors. In many cases, the physical aspects of caregiving were demanding, including lifting, bathing, dressing, toileting, and dealing with incontinence. One wife was “black and blue” having been hit by her husband when he didn’t understand why she was washing his genitals (Participant 2, Wife). Another wife fell and injured herself when trying to help her husband walk (Participant 3, Wife). However, caregivers did not dwell on the physical demands of their role and focused instead on other aspects of health and well-being:

#### 3.2.1. Time to Oneself and Ability to Prioritize Self

Caregivers associated being healthy with having the time and freedom to do what they wanted when they wanted to do it.

*It means having a little time to myself. It means not being a prisoner. It’s a massage, a pedicure. Being by yourself without having to organize it with places that will let me bring him [my husband]. I take him everywhere I go*.(Participant 3, Wife)

*Being able just to do whatever I wanted after school with my daughter and eating healthy meals and having a normal routine with my family. Just being able to have a normal life not in the dementia world*.(Participant 8, Daughter)

*Maintaining self-care time even if it is slim. Making some time for myself*.(Participant 7, Daughter)

#### 3.2.2. Positive Emotional State and Mental Well-Being

Caregivers also associated well-being with emotions like happiness and a positive state of mind. These emotions were often combined with the ability to self-prioritize.

*Well-being looks like happiness. Someone is always checking in on themselves and making sure they are in a good place. They make sure they are doing things that brings them joy, things for yourself*.(Participant. 1, Daughter)

*In the past I’d say well-being is a product of good health. But now I think good health is a state of mind, how you feel about your body, how you feel about your life*.(Participant 5, Husband)

*I think mental health is starting to become more of what I think about in terms of being healthy. It’s not just medical, physical health, but also mental health*.(Participant 3, Daughter)

#### 3.2.3. Social Engagement

A key aspect of health and well-being identified by the three youngest caregivers (age 38–48) was social engagement, especially the ability to interact with friends and family outside the home. These caregivers took socialization time for granted in the past but re-prioritized it as important to their health considering their changing responsibilities.

*You know, being able to socialize. I never would’ve thought of that as anything to do with well-being, but I do now*.(Participant 6, Daughter)

*It’s lunch with friends, dinner with friends. I would be able to make plans, meet friends or go somewhere with them right after work*.(Participant 7, Daughter)

*It’s just staying active, doing things outside of the home, staying socially engaged*.(Participant 8, Daughter)

#### 3.2.4. Spiritual Well-Being

Three of the older caregivers (age 52–83) indicated a spiritual element to health and well-being, noting that faith had been an important part of their lives prior to their caregiving duties and continued to play a key role. For some, spirituality was connected closely with a divine being; in other cases, it represented a connection to the community.

*Faith in Jesus Christ has always been a big part of our life. On the spiritual side [of well-being], it is spending some time every day with the Lord*.(Participant 2, Wife)

*I think there should be a spiritual element to it [health and wellbeing]. We have been faithful church goers and I miss that. I miss keeping in touch with the spiritual side—just being home, looking at videos has made me miss the church community*.(Participant 3, Wife)

*I believe there’s a God that loves us and cares for us more intimately and passionately than we can even imagine. And he wants us to care for ourselves too*.(Participant 4, Wife)

### 3.3. Barriers to Health and Well-Being

Most participants (N = 7) stated that their personal health and well-being had declined since being in their caregiver role. (The exception was Participant 4, a daughter, who had cared for her father for 2.5 years, far less time than other participants, who flew in to provide care for intermittent periods of time, and whose father was at a relatively early stage of Alzheimer’s disease compared to the patients of other caregivers). The remaining seven caregivers expressed an inability to make their own health and well-being a priority, even though they understood the importance intellectually. They characterized the subjective burden of the caregiving role as a mantle of responsibility that could not be shed. This mantle of responsibility was shaped by a confluence of emotions including grief, loss, resentment, isolation, and loneliness, all of which led to a lack of agency. As cited in other studies, these factors negatively impact caregivers’ mental health, health behaviors, and overall well-being [4,21].

*People have said “Now you gotta take care of yourself”. And you know, if you’re a care giver, you’re taking care of someone else, you’re not taking care of yourself. One of the reasons why is there is so much pressure to it. I try to continue things like Tai Chi and walking, but I’m still not, what would be the word, taking care of myself. I’m just trying to balance things with taking care of somebody else*.(Participant 5, Husband)

*Even when my mom lived in Nashville in her own condo, [or stayed with my brother on the East Coast], I felt responsible for everything*.(Participant 7, Daughter)

*People said, “you need to take care of yourself”. I said, “I can’t. I didn’t even feel like it was a choice—I have to take care of my mom, it has to be me*.(Participant 1, Daughter)

#### 3.3.1. Anticipatory Grief and Accumulated Loss

Participants’ experiences aligned with the concept of anticipatory grief. Anticipatory grief is an emotional response to an expected and inevitable loss and is separate from the grief associated with death [22]. Family caregivers of Alzheimer’s patients appeared to experience anticipatory grief as the person they loved changed, becoming psychologically absent while remaining physically present. This grief is exacerbated by complex and continuous loss that occurs at many levels and accumulates over time [21,23,24].

At one level, participants reported losses that occurred because of the profound loss of the person the caregivers once knew.

*My dad was an aeronautical engineer. He went from being an extremely bright and capable man to not being able to close an Excel spreadsheet. As he slowly deteriorates, it’s like, his sense of humor isn’t the same. And so, you are just constantly losing part of him*.(Participant 4, Daughter)

*[Being a caregiver] took a toll on my emotional health because being with somebody who is no longer the person that you really loved and spending a lot of time with the new person is really hard. And, you know, day after day, week after week, year after year, you kind of start to forget who the old person was and you’re just kind of presented with this new person. At times they can be like your old parent but it’s brief and it’s fleeting and then it’s gone*.(Participant 7, Daughter)

*Sometimes what they’re doing is not them. It’s what their problem is, but it creates this incredible feeling of loss, like when my wife says she can’t take it anymore and what did she do wrong to deserve this. You suffer constantly as the partner. If you are in love with someone and they don’t have a reason to live, you have to deal with that and you have to deal with that on your own*.(Participant 5, Husband)

At a second level, participants’ narratives conveyed profound change in the relationship between caregiver and care receiver. Caregiving has a relational dimension that cannot be separated from the past experiences and roles that caregiver and care receiver historically shared [25]. Daughters caring for mothers described experiencing a role reversal in which they were now the parent, tending to every need and keeping their mother safe. For some participants (N = 3), this parental caregiving occurred simultaneously with caring for young children.

*My mom got sick a year after my father died. I was going through a grieving process for that and then I went into a grieving process for her. It has been a lot of grief. The roles change—they become the child and you become the parent—it’s very confronting*.(Participant 1, Daughter)

*I wish I had my mother’s help [when I had my baby]. It would have been nice to have had her help with things and ask her questions. I wasn’t able to be home and just take care of my baby. I would sit on the couch at night and watch my mom on camera at her house, to see what she was doing and make sure she was okay, because the caregivers were a little iffy. At the same time, I’d be listening for the baby monitor*.(Participant 8, Daughter)

*I take responsibility for everything, you know. My mother is no longer able to do anything for herself. She does not recognize us. She cannot speak, she cannot walk, she cannot do anything. So, she’s total, 24-hour care. It’s not just the caretaking that is hard. There’s a tremendous amount of grief that you shoulder while you’re giving care. And that’s the part that’s the hardest*.(Participant 6, Daughter)

Caregivers of a spouse experienced a similar loss [26]. As the marital relationship lost it reciprocity and becomes more centered on the care recipient, participants expressed that they no longer experienced the same reciprocity in the relationship as they lost love and attention from their spouses.

*Before my husband was diagnosed with Alzheimer’s, we were very active. We socialized. We were part of a couple’s group for potluck dinners. Now he goes to bed at 6 or 6:30 p.m. He just collapses, like a balloon that is pricked. Our friends are gone. We are no longer part of the same community. For me there is a lot of alone time*.(Participant 3, Wife)

At a third level, there is loss of self. This is the loss of identity that comes from being immersed in the caregiver role, with limited access to social contact or a social role outside that of caregiver. Loss of self is associated with lower levels of self-esteem and mastery [27], further contributing to the caregivers’ lack of ability to prioritize their own health and well-being.

*I don’t have the ability to socialize or go out because I don’t want to leave my mother alone. I just feel like you put yourself on the back burner*.(Participant 1, Daughter)

#### 3.3.2. Regret and Guilt

While the specific word “regret” was not used by caregivers, all participants interviewed expressed some type of regret or yearning for the life they lived before becoming a caregiver. They acknowledged missing the simple things they had taken for granted like “just being able to run to the store” (Participant 8, Daughter), “being able to go to the doctor” (Participant 6, Daughter), or “going outside for a walk” (Participant 2, Wife).

Caregivers’ feelings of love and empathy for their loved ones were often mixed with feelings of resentment, followed by guilt for feeling the resentment.

*My mental well-being took a hit. I was also very resentful, having to constantly care for my mother, clean up after her, and deal with her mood changes. I did not exercise, eat well, or sleep well during that time. I didn’t have the ability to care for myself except walk my dog. There was no exercise and we just ordered take out, not the best diet. I would say 100% that I was depressed during that time. I had just lost my father and I was losing my mom at that same time*.(Participant 1, Daughter)

*If I had the choice of like going and doing something for myself or picking my mother up, I would just kind of swallow the lump in my throat and would pretty much go pick her up. I was definitely with her more than I wanted to be. Which sounds horrible. She’s my mother and I should want to be with her all the time. But you don’t really want to be with them, you know you don’t, you’d rather do anything else. You know, it’s like putting your hand on a hot stove. You know it’s going to hurt you, but you do it anyway. And when you do it long enough it, it really just starts to affect you mentally*.(Participant 7, Daughter)

Caregivers also acknowledged guilt just for having time to themselves. They felt guilty if they even had an hour to enjoy a small aspect of the life they used to have.

*On Monday I had a caregiver—I didn’t plan anything—so my neighbor and I went out to eat. But I felt guilty. I could have gone to the grocery store and because I didn’t, we didn’t have anything to eat. I just can’t get away*.(Participant 2, Wife)

Caregivers also had the capacity to acknowledge the positive in their lives, rather than simply the regret for the life they had and shared with their loved one, provided they made a conscious effort to do so. In both examples below, the participants also identified spirituality as an important part of well-being.

*This is where the mental health piece comes in. I have to be mindful of how I’m feeling and what I’m thinking. And, you know, when I’m feeling down and sad about my situation, I can remind myself of all the good things in my life. All the great things, the blessings*.(Participant 6, Daughter)

*Sometimes I feel like I can’t do this, it’s too hard. This opens the door to self-pity. But I am stronger than that. The Lord will give me what I need*.(Participant 2, Wife)

#### 3.3.3. Isolation and Loneliness

All but one caregiver (N = 7) expressed feelings of isolation and loneliness that acted as a barrier to their health and well-being. They described isolation from everyday social activities that they used to participate in, such as attending church, enjoying a meal with friends, or doing a workout at the gym. The reasons for this isolation varied from having to be home because “no one else can watch them” to losing touch with friends with whom they no longer shared things in common. In some cases, loneliness was paradoxical, as the caregivers were both lonely and never alone, needing to take their loved one with them to every activity outside of the house. Isolation and loneliness further reinforce loss of self for the caregiver.

*I can’t not be here. I can’t just go out, you know. If I want to go somewhere I have to make sure somebody’s here, but I can’t find people I trust. My husband is great, but he comes, and he goes, so sometimes I’m here by myself with my mother for the whole weekend. Those times are so hard*.(Participant 6, Daughter)

*My husband is now spending a lot of time in bed. He doesn’t want to get up and get dressed. So, unless I can get someone to come over and be here, I can’t go anyplace or do anything. I feel like a prisoner. It’s not a 12 × 12 room but it is solitary confinement*.(Participant 3, Wife)

*If I went out, I brought my mother with me because I didn’t want to leave her alone. I was never able to have any alone time for myself. I couldn’t even watch TV downstairs by myself. Besides going to work, I didn’t go anywhere without her because I didn’t feel comfortable*.(Participant 1, Daughter)

#### 3.3.4. Lack of Agency

Agency is the feeling of control that one has over their actions and the related consequences [28]. A sense of agency is important because it is linked to the motivation to take action and regulate health behaviors [29,30]. In these interviews, participants expressed a lack of agency which may or may not have reflected the objective reality of their situation. Lack of agency was expressed in several ways, such as lack of motivation to engage in health behaviors, lack of physical or mental energy to perform tasks outside the caregiving role, and a strong belief that their caregiving role could not be delegated to someone else.

*I used to be good [at scheduling my doctor’s appointments], but it’s not natural now. And I think that I used to schedule appointments after work, which I can’t do now because I am with my mother every evening. So, I have to take off work, which is not as easy to do, to be able to go to the doctor’s appointment. I definitely procrastinate. I’d probably have to be in a lot of pain in order for me to pay attention to something, which is not good, which is what happened to me about a year and half ago*.(Participant 6, Daughter)

*[After a full day of work], I would, would be at my mother’s home for hours until I laid her down for bed. And then I would finally head home for the night and I would literally walk in the door crying just because I was so drained. I was completely, mentally shot. So I would get home to my house. But who wants to do anything when you feel like that? But then my night wasn’t over because I would keep the camera on, on my phone, and I would stare [at the cameras we installed in her bedroom] and make sure she was asleep*.(Participant 8, Daughter)


*I think of myself as one person my wife could trust, and I will take that to the grave—who else can she trust? Who else is going to be there and a closeness that I share with this person?*
(Participant 5, Husband)

#### 3.3.5. Impact of Barriers as Reported by Caregivers

When asked to compare their health and well-being before and after becoming a family caregiver, most caregivers (N = 7) reported that their health and well-being had declined significantly. They reported poorer health, including less healthy eating habits, less exercise, poorer sleep, and fewer doctors’ appointments as a result of their caregiving responsibilities. The one exception was a participant whose father was still in the early stages of Alzheimer’s disease and who was still able to live independently. For this participant, her father’s illness was a motivator to take better care of herself to be available for him, and this self-care message was strongly reinforced by an Alzheimer’s support group she attended.

The remaining participants attributed the cumulative effect of grief, depression, stress, lack of time, and lack of motivation to the decline in their health behaviors.

*The grief that I was experiencing after my mom moved was from the letdown of what I had been going through for the last ten years, combined with COVID. I was drinking like, I mean, not enough to say you’re an alcoholic or something, but for me, way more than I ever had. And that led me to just eat, you know, more unhealthy food. You know, you drink and then you eat badly, then you just feel gross and then you eat more gross food, and the whole thing just gets out of whack. Then your sleep starts to be disrupted*.(Participant 7, Daughter)

*I mean, obviously I wasn’t eating right because I was always so busy. I was just picking at whatever was left over from the baby’s food and my mom’s food*.(Participant 8, Daughter)

### 3.4. Strategies to Support Health and Well-Being

Participants discussed strategies to support their own health and well-being from both the perspective of the objective burden of care (e.g., more time away from the caregiving role and more money to support this) and the subjective burden of care (e.g., caregivers’ perception of how the caregiver role impacted their physical, emotional, social, and spiritual health). While some participants (N = 5) mentioned “more time and/or money” as being helpful, it is important to note that this solution alone would not alleviate the subjective burden of care which was previously described as “a mantle of responsibility that could not easily be shed”. For example, half the participants (N = 4) felt that they were “the only one who could really take care of” their loved one, or that the full responsibility fell on their shoulders (even if other support was available). Participants also expressed guilt for being away from their loved one, especially if it was for a pleasurable activity like lunch with a friend versus a productive errand like grocery shopping. Participants experienced an overwhelming amount of mental and physical exhaustion and lack of motivation to participate in health-promoting behaviors, even if time was available. In some cases, participants recognized the importance of time away from an intellectual perspective but could not reconcile it with their current situation.

*Maybe the thing that could help caregivers the most is getting a real two or three hours away. If they could walk out the door two or three times a week and think “Mom’s okay. Dad’s okay. My wife is okay”. That would energize the brain. In my situation now, that is ridiculous*.(Participant 5, Husband)

Thus, in our interviews, we probed for strategies to address the subjective burden of care. One finding was surprising and became an organizing theme for these strategies: participants were largely ignored by healthcare professionals, with virtually no attention paid to caregiver health and well-being over the years they cared for their family member with Alzheimer’s.

#### 3.4.1. Recognition of Caregivers as “Additional Patients” Not “Invisible Patients”

The process of diagnosing a patient with Alzheimer’s or other forms of dementia is arduous and it may take years to reach a diagnosis [31]. Understandably, the neurologist or other healthcare professional who finalizes this diagnosis is focused on the Alzheimer’s patient, not the caregiver. However, the caregivers interviewed for this study recognized the importance of receiving some type of support at this point, early on in their caregiving journey. Only two of the eight participants in in this study received a referral to an Alzheimer’s support group from their healthcare professional at time of diagnosis. These two participants found the combination of general information, classes, and personal support to be “useful” and, in one case, “phenomenal”. For the six other participants, no guidance was provided, leaving them to navigate the caregiving journey and find support services on their own, along with the myriad of other care responsibilities.

*To be honest, there just wasn’t anything. I mean, never at one single neurology appointment was any care or help offered to me. Nothing was ever shared from an education standpoint. It was really always just, ‘well, your mom had this and we’ll see you in six months, or you let us know if you have any questions or concerns about her medication’*.(Participant 7, Daughter)

Beyond a lack of information, there was a lack of interest expressed in their well-being as caregivers.

*“I think that you [Interviewer SD] are the first person that has asked me how I’m doing related to my mom and taking care of her. No, I’m not talking about friends, because I have friends—but like, in any kind of a professional capacity. My mom’s doctors are nice, and they make house calls, which is phenomenal. Her nurse practitioner comes here to see her, which is great. And they’re very kind. But there’s nothing they can do for me*.(Participant 6, Daughter)

*“When I got your invitation to be in this research I thought, ‘Why not? “I get to talk to an adult during the day”’*.(Participant 3, Wife)

#### 3.4.2. Roadmap Facilitated by a Dementia Case Manager

Participants who had not been directed to support services when their family member was first diagnosed expressed strong interest in having some type of roadmap to guide them through the disease progression and provide concrete, practical advice to help them navigate the financial, insurance, and legal challenges associated with caring for someone with dementia. While these participants didn’t articulate the need for a “dementia case manager” [32], they provided a description of this role in their interview commentary.

*I wish there was an offering of some type of counselor who could come in and check on me. When my mother was in hospice, someone from the clergy would come to check on her, and she would always check on me too. You know, it was nice—at least someone was asking me, “how are you doing?” So maybe for the person who has the disease, there is some type of care plan for the caregiver, some kind of resource that’s free*.(Participant 6, Daughter)

*There’s not an instruction manual on the journey but it would have been good to have some guidance and someone to help facilitate this journey. Like after the neurologist diagnosis if someone said, here are local resources in your area, here’s where you go for help with power of attorney, here’s how to get help with in-home care. Anything like that—but you just have to figure this out on your own. Instead, we were just told the medications to take. There was no follow up. I didn’t feel like I had anyone to turn to. There needs to be a more social services end of it, someone who would have a better idea of what the patient and I would go through*.(Participant 1, Daughter)

#### 3.4.3. Non-Traditional Support Tailored for Specific Life Stages and Challenges

Of the eight participants in this study, the only two who had participated in any type of support activities for Alzheimer’s caregivers were those who had been referred to these support groups at the time of diagnosis. The other participants were asked what type of support services might be most valuable, even though they hadn’t formally participated. Key findings included (1) support services that allowed joint participation with the care recipient and (2) fun, unstructured activities with other caregivers, such as a harbor cruise and lunch mentioned by one participant.

*In the desert, our local Alzheimer’s Association was excellent. My husband and I went twice a week to Club Journey. There were chair exercises, then cards, and dominos with other caregivers and loved ones, then lunch. The time was about 10 a.m. to 1 p.m. My husband loved it. He would get up and say, “is it Tuesday?’ because that’s when we had Club Journey*.(Participant 3, Wife)

Participants expressed two key reasons they did not participate in support activities. The first was lack of fit for their specific needs and this life stage. Younger caregivers whose mothers had been diagnosed with earlier onset Alzheimer’s expressed an inability to identify with members of the more traditional Alzheimer’s support groups.

*I went to one in-person support group. It was me and two other people, but they were women in their 60s and both of their spouses were dealing with Alzheimer’s. Being a caregiver for your parent is different from your spouse. Alzheimer’s typically hits someone older, so it was hard to find a specific group*.(Participant 1, Daughter)

*There wasn’t anything for me, even at the Alzheimer’s Association. I felt like a spring chicken in a room full of dinosaurs. I mean, I was so young [late 20s]. I finally found help through the Young Professional Board in Nashville. It’s just a group of people under 40 that, you know, provide support, although it was a lot more about raising money. But I was just so grateful to find anyone under the age of 40 that had any idea what I was going through*.(Participant 7, Daughter)

There was a bright spot in Participant 7’s unsuccessful search for an Alzheimer’s support group. She went on to create a non-profit that provides free respite care to family caregivers of Alzheimer’s patients.

A second reason cited for lack of participation in a support group was lack of time, energy, and motivation to seek out support services.

*I searched for Alzheimer’s support groups, and I went to the Alzheimer’s Association website but even that was taxing. If you have any free time you don’t want to be researching more Alzheimer’s*.(Participant 1, Daughter)

*So they have the Alzheimer’s or whatever association, I don’t know what they’re called, but there’s a chapter or something here in my county. Like, I reached out to them in the beginning when my mother first got diagnosed and I got a packet. But I haven’t really utilized them for anything since then, because I just used what I had at my disposal*.(Participant 6, Daughter)

## 4. Discussion

The purpose of this study was to better understand barriers to health and well-being from the unique perspective of informal caregivers of patients with Alzheimer’s and other forms of dementia, and to identify strategies to help these caregivers better prioritize their own health and well-being based on their lived experiences. While acknowledging the objective burden of care (the time, activities, and finances devoted to care), our research focused on the subjective burden (the caregivers’ perception of the impact of their caregiving role to their physical, emotional, social, and spiritual health) that has been largely under-represented in the literature [10]. This study focused on the caregivers’ own definition of “health and well-being”; barriers that they perceived to prioritizing their own health and well-being, and the impact of these barriers; and strategies that would help them and other caregivers in similar roles take better care of their health.

### 4.1. Definitions of Health and Well-Being

Overall, participants’ definitions of health behaviors were consistent with those in similar studies of informal caregivers of Alzheimer’s patients. In our study, definitions included healthy eating, exercise, sufficient sleep, and making time for medical appointments. Most participants noted that these behaviors declined because of their caregiver role. These findings are similar to those cited in a qualitative study by Wang et al. [4], which cited lack of healthy eating, a severe lack of sleep, lack of physical activity, and deferring medical appointments as “neglected self-care” that was negatively influenced by the caregiving role. While Wang et al. [4] concluded that the neglected physical self-care jeopardized mental and social self-care, participants in our study provided a more cursory mention of physical self-care and instead emphasized positive mental health, social engagement, and, in some cases, spiritual health in their definitions of “health and well-being”.

### 4.2. Barriers to Health and Well-Being

Our study also focused on the barriers to prioritizing caregivers’ health and well-being and found that a confluence of emotions, including grief, loss, regret, guilt, loneliness, and isolation, lead to a lack of agency or inability to control their own actions and consequences [28]. A sense of agency is important with respect to health because it is linked to the motivation to act and regulate health behaviors [29,30]. For our participants, lack of agency was observed as a lack of motivation to engage in health behaviors, a lack of energy to perform tasks outside the caregiving role, and a strong belief that their caregiving responsibilities could not be delegated to someone else. These complex emotions and lack of agency became a “mantle of responsibility” that could not easily be shed, even if other caregiving resources were available.

Other studies cite similar psychological barriers to well-being in family caregivers of Alzheimer’s patients, including stress, loss, grief, diminished sense of self, yearning for the past, isolation, restricted freedom, yearning for normalcy, and lower self-efficacy [3,4,21,23,33]. Notably, our research focused on the inter-relationship between accumulated loss and anticipatory grief, that is under-studied in the literature about informal caregivers of Alzheimer’s patients. Loss and grief are inexplicably linked, as grief is a universal human reaction to loss, whether that loss is actual or perceived [34]. Accumulated loss refers to the broad spectrum of losses that occur during the long progression of Alzheimer’s, including the loss of a person who is physically present but psychologically absent, the loss of anticipated or reciprocal relationships, and the loss of self, as one’s own identify becomes increasingly narrowed by the caregiver role [22,34].

By contrast, anticipatory grief, sometimes referred to a pre-death grief, is an “emotional response to an expected and inevitable loss which occurs before the actual loss” [22]. In this way, anticipatory loss is different than the mere anticipation of death or the grief one experiences when a loved one passes away, which caregivers will also experience in the future. Anticipatory grief is associated with many of the other emotions that were noted as barriers to health and well-being, including loneliness, isolation, yearning, diminished sense of self, lack of freedom, anger, profound sadness, depression, and anxiety [21,23]. Thus, it is possible that accumulated loss and anticipatory grief are drivers of other negative emotions that interfere with the physical health and mental well-being of informal caregivers of Alzheimer’s patients, although more research is needed to understand this complex relationship. The available research on anticipatory grief shows that it may not be recognized by the caregiver or by health professionals [23,35]. If ambiguous grief is unresolved, it may lead to complicated grief after the loved one passes. In cases of complicated grief, the feelings of loss are debilitating and don’t improve over time, preventing the individual from recovering from the loss and resuming their normal life [24].

### 4.3. Strategies to Improve Health and Well-Being

The primary strategies caregivers identified to improve their health and well-being addressed both the objective burden and the subjective burden of care. Most recipients indicated that more respite care and the financial support to find this would improve their well-being; this is consistent with other relevant studies [4,36]. However, we noted that more time away and the finances to support it did not necessarily address the subjective burden of care (the guilt, lack of motivation, overwhelming senses of responsibility, and extreme mental and physical exhaustion experienced by caregivers). In many cases, participants were unable to relinquish their caregiver role, or at least the mental responsibility of it, even if other caregiving options were available. This is consistent with the findings of Brodaty and Dunkin [3], who showed that subjective burden is only loosely coordinated with objective burden.

With respect to the subjective burden experienced by caregivers, our research cited strategies that have implications for healthcare professionals, governmental agencies, non-profits, and businesses that provide support services to caregivers. These strategies included the recognition of caregivers as “additional patients” not “invisible patients” [3] by healthcare providers and support service referral at the time a patient is diagnosed with Alzheimer’s or other form of dementia. The second strategy identified is a roadmap of what to expect as the disease progresses, managed by a dementia case manager: a designated healthcare or social worker who can help the caregiver navigate the financial, legal, insurance, home care, day care, and self-care challenges throughout the disease progression. Messina et al. [32] cite the importance of a dementia case manager to facilitate access to local support services and interventions that are geared to caregiver needs and expectations. Additional research is needed to understand how this type of service may be implemented and systematically evaluated, potentially in conjunction with family physicians and other healthcare services.

The third strategy to improve the health and well-being of family caregivers was non-traditional support tailored for specific life stages and challenges. Most of the participants in our study had not taken advantage of support services due to the overwhelming responsibilities of their role and lack of fit for the challenges they were facing at their specific life stage. Similar studies show a general reluctance of caregivers to seek help [32], demonstrating that the most successful interventions are those that are tailored to the needs of the individual and specifically address the subjective burden they experience [3]. There is a growing need for support services tailored to younger family caregivers, since Alzheimer’s disease, which largely affects older adults, increasingly affects younger adults aged 30 to 64 [37,38]. This uptick in earlier onset Alzheimer’s increases the likelihood that younger caregivers in their twenties, thirties, and forties will navigate care for a parent while simultaneously caring for their children. These caregivers are also more likely to be working, have greater financial responsibilities, and provide care for a longer period than other caregivers [3].

### 4.4. Limitations

It is important to note some limitations in this research. First, our sample size is small, and the findings are not intended to be representative of the experiences of all caregivers. However, our sample was heterogenous in nature and represented participants in a wide range of ages (32–83), geographies (across the US), and caregiving roles (daughters, wives, and one husband). Second, the caregiving experiences described by participants all included the COVID-19 period. The pandemic, shelter-in-place requirements, and social distancing most likely contributed to the physical and mental health of the caregivers, beyond their caregiving role, but further research would be required to identify this impact (which is true of any research conducted during the pandemic). Third, the interview protocol did not include questions that could elicit potentially maladaptive strategies (e.g., substance use). Thus, the findings from this study are limited by what participants were willing to disclose. Finally, our research did not explore differences in the subjective level of burden experienced by caregivers based on length of time in their role or the progression of Alzheimer’s experienced by their loved one (mild, moderate, severe). The literature shows mixed results with respect to the impact of these two factors [4,24], and further research is required.

## 5. Conclusions

The subjective burden of strain experienced by family caregivers of Alzheimer’s patients has a profound impact on their health and well-being, even more so than the objective burden of strain that is the result of their day-to-day caregiving activities. Participants in this study reported a sense of subjective burden, or the perceived psychological impact of their role, that is shaped by grief, loss, regret, guilt, loneliness, social isolation, and lack of agency. The confluence of these emotions is like a heavy mantle that, once donned, is hard to remove, even if circumstances change.

Future consideration should be given to specific support services that can help address the anticipatory grief associated with dementia caregiving, which is an area with minimal research. This area of research should also include comparison with formal caregivers to identify generic and specific stressors, health needs, and coping styles for varied approaches to social care.

More work in this area can enable mindset shifts on the part of healthcare professionals, social workers, governmental agencies, and nonprofits so that they view these caregivers as “additional patients” with distinct needs related to their physical, emotional, and social well-being.

## Figures and Tables

**Table 1 ijerph-20-04328-t001:** Themes and sub-themes.

Interview Question	Theme	Sub-Themes
Defining health and well-being	Mental well-being and social engagement are prioritized over physical health or health behaviors	Time to oneself and ability to prioritize self Positive emotional state and mental well-being Social engagement Spiritual well-being
Barriers to health and well-being	The subjective burden of caregiving creates a mantle of responsibility that cannot easily be shed.	Anticipatory grief and accumulated loss Regret and guilt Isolation and loneliness Lack of agency
Strategies to promote health and well-being	Recognition of caregivers as “additional patients” with appropriate, tailored support	Recognition of caregivers as “additional patients” not “invisible patients” Roadmap facilitated by dementia case manager Non-traditional support groups tailored for specific life stages and challenges

**Table 2 ijerph-20-04328-t002:** Sociodemographic characteristics of participants.

ID	Age	Gender	Work Status	Family Status	Patient Age	Relationship to Patient	# of Caregiving Years	Caregiving Location
1	32	F	Work FT	Single, No Children	61	Daughter	3	Caregiver’s home
2	74	F	Retired	Married, 4 Children	80	Wife	6	Caregiver’s home, then Memory Care facility
3	83	F	Retired	Married, 4 Children	82	Wife	5	Caregiver’s home
4	52	F	Work FT	Married, 1 Child	81	Daughter	2.5	Father’s home
5	80	M	Retired	Married, 2 Children	81	Wife	10	Caregiver’s home
6	48	F	Work FT	Married, 3 Children	83	Daughter	8	Caregiver’s home
7	42	F	Work FT	Married, 3 Children	72	Daughter	12	Mother’s home, Brother’s home, Caregiver’s home, Assisted Living
8	38	F	Work FT	Married, 1 Child	70 (deceased)	Daughter	6	Mother’s home

Abbreviations: FT = Full Time.

## Data Availability

The data presented in this study are available upon request from the corresponding author. The data are not publicly available due to privacy restrictions.

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
