# Peer review of "Barriers and Facilitators of Health and Well-Being in Informal Caregivers of Dementia Patients: A Qualitative Study"

_ijerph, 2023, doi:10.3390/ijerph20054328_

Round 1

Reviewer 1 Report

The authors might want to consider revising the following:

1)     Conclusion section should be revised. Conclusions should be based on the obtained data. They should not, just because of the small sample size, amount to wishful thinking or suggesting implementation of certain solutions.

Author Response

Reviewer

Comment

Revision

R1

Conclusion section should be revised. Conclusions should be based on the obtained data. They should not, just because of the small sample size, amount to wishful thinking or suggesting implementation of certain solutions.

We appreciate this suggestion. We have removed recommendations for future practice and replaced it with a focus on the need for more research in this area in the conclusion. We also modified the conclusion in the abstract accordingly.

Reviewer 2 Report

Dear authors, Thank you for let me review your manuscript titled "Barriers and Facilitators of Health and Well-Being in Informal Caregivers of Dementia Patients: A Qualitative Study.

In my opinion it is a relevant topic and the manuscript has a role to drew attention about it.

Only some considerations and suggestions:

In methodology I wonder how authors managed the "saturation criteria".

In discussion I miss some comparison with "formal caregivers"  feelings who care dementia patients in the same home. Or pointed out as future research. 

And about health and well-being, some questions about drug consumption to cope depression, insomnia, etc...

Congratulations to authors, 

Author Response

R2

In methodology I wonder how authors managed the "saturation criteria".

Thank you for this comment. Contemporary methodological literature about quality in qualitative research has expanded criteria for achieving rigor in the analysis process. Instead of saturation, we used the criteria of heterogeneity, which is coherent with the interpretivist paradigm and reflexive thematic analysis. We have added a sentence about the heterogeneity in the Data Analysis section: “Through this cycle, we attempted to achieve the research quality criteria of heterogeneity by seeking to illuminate variation and distinctiveness within and across participants’ reports of their experiences [14].”

R2

In discussion I miss some comparison with "formal caregivers"  feelings who care dementia patients in the same home. Or pointed out as future research. 

Thank you for this suggestion. We have added this to the Conclusion section to point to areas for future work.

R2

And about health and well-being, some questions about drug consumption to cope depression, insomnia, etc...

Thank you for this comment. Since the interview protocol did not include questions related to maladaptive coping strategies, we added a sentence to the Limitations to note this: “Third, the interview protocol did not include questions that could elicit potentially maladaptive strategies (e.g., substance use). Thus, the findings from this study are limited by what participants were willing to disclose.”

Reviewer 3 Report

Congratulations on your hard work in writing this manuscript.

It is a very interesting article.

I propose to the author to provide some details regarding the process of developing and validating the questionnaire used to conduct the semi-structured interview of the subjects.

As a reader, I would like to find out how many questions the questionnaire used to interview the respondents has.

The first section of the article (Introduction) is designed accordingly.
I recommend that at the end of the introduction, the objective of the study should be clearly formulated.

- In the Methodology section, I recommend adding additional information about the data collection tool. It is important to provide details regarding the number of questions, validation methods of the instrument with which the interview was carried out, elements of originality or if it is an instrument taken from specialized literature.

- We recommend the formulation of additional clarifications in relation to the sampling process for determining the number of subjects included in the study.

- According to the recommendations from the specialized literature, the sampling project in the qualitative study must specify minimum samples, based on the reasonable coverage of the phenomenon depending on the purpose of the study and the interests of the interested parties.

- I recommend the authors to specify what measures they used to limit the phenomenon of bias.

- Additions are needed regarding the ethical aspects of the research: data confidentiality, maintaining the anonymity of the subjects included in the study, signing the consent, Informing participants about the results, respect for privacy, reporting how the sample was selected.etc.

- In the discussion section I propose to add some suggestions regarding further research on the same topic.

- In the table it is necessary to add the values of the standard deviation and the average age for the subjects included in the study

Another  positive observations:

- The references cited are predominantly recent and relevant publications and do not include excessively self-cited material.

- The final statements and conclusions are consistent and supported by the listed quotations.

- Considering that the study is of a qualitative type, the tables are consistent with the specifics of this type of study.

Author Response

R3

I propose to the author to provide some details regarding the process of developing and validating the questionnaire used to conduct the semi-structured interview of the subjects.

As a reader, I would like to find out how many questions the questionnaire used to interview the respondents has.

Thank you for this suggestion. We have added more details about the interview protocol pilot, including example interview questions.

R3

The first section of the article (Introduction) is designed accordingly.
I recommend that at the end of the introduction, the objective of the study should be clearly formulated.

Thank you for this suggestion. We have added “the purpose of this study was” to a sentence in the introduction to address this.

R3

In the Methodology section, I recommend adding additional information about the data collection tool. It is important to provide details regarding the number of questions, validation methods of the instrument with which the interview was carried out, elements of originality or if it is an instrument taken from specialized literature.

Thank you for this suggestion. We have added more details about the interview protocol pilot, including example interview questions.

R3

We recommend the formulation of additional clarifications in relation to the sampling process for determining the number of subjects included in the study.

Thank you for this comment. Contemporary methodological literature recommends using the quality criteria of heterogeneity, rather than a specified sample size, to determine whether a study rigorously and thoroughly described the experiences of research participants. To achieve this, we have presented patterns with attention to variation between participants’ experiences and, where appropriate, instances when a participant’s experience differs from the general pattern. We have included a sentence about the principle of heterogeneity to the Data Analysis section.

R3

According to the recommendations from the specialized literature, the sampling project in the qualitative study must specify minimum samples, based on the reasonable coverage of the phenomenon depending on the purpose of the study and the interests of the interested parties.

See the comment above.

R3

I recommend the authors to specify what measures they used to limit the phenomenon of bias.

Thank you for this comment. The manuscript includes a Reflexivity/Positionality statement to account for the role of researcher subjectivity in the research bias. In that section, we added a sentence about seeking instances that did not align with our expectations. This also relates to the principle of heterogeneity for quality in qualitative research. 

R3

Additions are needed regarding the ethical aspects of the research: data confidentiality, maintaining the anonymity of the subjects included in the study, signing the consent, Informing participants about the results, respect for privacy, reporting how the sample was selected, etc.

Thank you for this comment. We added additional details about required procedures for the Institutional Review Board’s requirements for ethical research under the Exempt category.

R3

In the discussion section I propose to add some suggestions regarding further research on the same topic.

Thank you for this comment. We have added additional reference to more research needed in the Conclusion section in addition to what is noted in the Discussion section.

R3

In the table it is necessary to add the values of the standard deviation and the average age for the subjects included in the study

Thank you for recommendation. Since the analytical focus of this paper is themes across participants reported experiences, we have elected not to calculate or make statistical claims related to participants’ ages or quantitative descriptors. These details are provided for readers as context about participants’ experiences.